# Hyporesponsiveness to erythropoiesis-stimulating agent in non-dialysis-dependent CKD patients: The BRIGHTEN study

**Ichiei Narita** *, **Terumasa Hayashi, Shoichi Maruyama, Takao Masaki, Masaomi Nangaku, Tomoya Nishino, Hiroshi Sato, Tadashi Sofue, Takashi Wada, Enyu Imai, Manabu Iwasaki, Kyoichi Mizuno, Hiroki Hase, Masahiro Kamouchi, Hiroyasu Yamamoto, Tatsuo Kagimura, Kenichiro Tanabe , Hideki Kato, Takehiko Wada, Tomoko Usui, Tadao Akizawa, Hideki Hirakata, Yoshiharu Tsubakihara**

Division of Clinical Nephrology and Rheumatology, Niigata University Graduate School of Medical and Dental Sciences, Niigata, Japan

* naritai@med.niigata-u.ac.jp

**Data Availability Statement:** All relevant data are within the paper and its Supporting information files.

## Abstract

Among non-dialysis-dependent chronic kidney disease (ND-CKD) patients, a low hematopoietic response to erythropoiesis-stimulating agents (ESAs) is a predictor for poor renal and cardiovascular outcome. To assess the method for evaluating hyporesponsiveness to ESA in patients with ND-CKD, a multicenter, prospective, observational study of 1,980 adult patients with ND-CKD with renal anemia was conducted. Darbepoetin alfa (DA) and iron supplement administrations were provided according to the recommendation of the attached document and the guidelines of JSDT (Japanese Society of Dialysis and Transplantation). The primary outcomes were progression of renal dysfunction and major adverse cardiovascular events. ESA responsiveness was assessed using pre-defined candidate formulae. During the mean follow-up period of 96 weeks, renal and cardiovascular disease (CVD) events occurred in 683 (39.6%) and 174 (10.1%) of 1,724 patients, respectively. Among pre-set candidate formulae, the one expressed by dividing the dose of DA by Hb level at the 12-week DA treatment was statistically significant in predicting renal (hazard ratio [HR], 1.449; 95% confidence interval [CI], 1.231–1.705; P<0.0001) and CVD events (HR, 1.719; 95% CI, 1.239–2.386; P = 0.0010). The optimum cut-off values for both events were close to 5.2. In conclusion, hyporesponsiveness to ESA in ND-CKD cases, which is associated with a risk for renal and CVD events, may be evaluated practicably as the dose of DA divided by the Hb level at the 12-week DA treatment, and the cut-off value of this index is 5.2. A search for the causes of poor response and measures for them should be recommended in such patients.

**Trial registration:** ClinicalTrials. gov Identifier: NCT02136563; UMIN Clinical Trial Registry Identifier: UMIN000013464.

**Funding:** Translational Research Center for Medical Innovation has received a research grant, which was not specific for this study from Kyowa Kirin Co., Ltd. (KK). Kyowa Kirin was not involved in designing, data interpretation, and manuscript writing for this study, along with any other matters including employment, consultancy, patents, products in development, marketed products, etc.

**Competing interests:** Dr Narita reported receiving lecture fee and grant from Kyowa Kirin (KK). Dr Hayashi reported receiving honoraria from KK. Dr Maruyama received honoraria and subsidies or donation from KK. Dr Masaki reported receiving lecture fee and grant from KK. Dr Nangaku reported receiving honoraria, manuscript fees, and subsidies or donations from KK. Dr Nishino reported receiving lecture fee and grant from KK. Dr Sofue reported receiving grant from KK. Dr Takashi Wada reported receiving honoraria and subsidies or donation from KK. Dr Hase reported receiving honoraria from KK. Dr Yamamoto reported receiving honoraria and manuscript fees from KK. Dr Takehiko Wada reported receiving honoraria from KK. Dr Akizawa received honoraria and manuscript fees from KK. Dr Tsubakihara received honoraria from KK. This does not alter our adherence to PLOS ONE policies on sharing data and materials.

## Introduction

Anemia is a common complication in both dialysis and non-dialysis-dependent chronic kidney disease (ND-CKD) patients [1–3]. Low levels of red blood cells and hemoglobin (Hb) are mainly attributed to the insufficient effect of erythropoietin (EPO), resulting from reduced EPO production by kidney cells and decreased response to EPO, which is attributed to several factors (i.e., underlying inflammation, comorbid type 2 diabetes or cancer, and iron deficiency) [4, 5]. The introduction of recombinant human erythropoietin (rHuEPO) into clinical practice in the 1980s was a significant breakthrough in the treatment of anemia in these patients [6, 7]. The use of erythropoiesis-stimulating agents (ESAs) has extensively improved morbidity, mortality, and kidney function and quality of life in these patients [8–11], although the target Hb level in patients with ND-CKD undergoing ESA treatment remains controversial [10, 12, 13]. Several large-scale randomized controlled trials, which compared the benefit of target Hb level as normal (>13 g/dL) versus lower (10–11 g/dL) on mortality and cardiovascular disease (CVD) events, have consistently shown increased risk and no incremental improvement in the quality of life of normalizing Hb levels [14–16]. These results may indicate that, for patients with CKD with hyporesponsiveness to ESA, increasing the ESA dose further is likely to increase risks for CVDs.

It is obvious that the ESA dosage required to achieve correction of anemia varies among patients with CKD, and hyporesponsiveness to ESAs has been well recognized as a strong predictor of poor renal and CVD events [10]. The 2012 Kidney Disease Improving Global Outcomes clinical guideline for anemia of CKD recommends evaluating patients with initial or acquired ESA hyporesponsiveness and balancing the potential benefits of treatment with intravenous iron or ESA therapy as prophylaxis against adverse reactions [17]. However, the appropriate formula to evaluate hyporesponsiveness to ESA in patients with ND-CKD remains to be established. It has recently been recognized that hypoxia-inducible factor-prolyl hydroxylase (HIF-PH) inhibitors may be particularly useful in patients with CKD with hyporesponsiveness to ESA [18–21]. However, owing to the uncertainty involved in defining ESA hyporesponsiveness in patients with ND-CKD, it is difficult to establish a guiding principle for conversion or choice between ESA and HIF-PH inhibitors [22].

BRIGHTEN (oBservational clinical Research in chronic kidney disease patients with renal anemia: renal proGnosis in patients with hyporesponsive anemia to ESAs, darbepoetiN alfa) is a multicenter, prospective observational study aimed at establishing an appropriate definition for hyporesponsiveness to ESAs that accurately predicts poor renal outcome and CVD events in patients with ND-CKD in a real-world clinical setting in Japan.

## Materials and methods

### Study design and patients

The design of BRIGHTEN, along with the inclusion and exclusion criteria, has been described in detail elsewhere [23, 24]. The study was designed, implemented, and overseen by the BRIGHTEN Executive Committee, together with representatives of the Translational Research Center for Medical Innovation, Kobe, Japan, a third-party organization independent of the investigators' institutions and responsible for data collection and analysis. It was conducted under the health insurance system of Japan and in accordance with the principles of the Declaration of Helsinki and the Ethical Guidelines on Clinical Studies of the Ministry of Health, Labor, and Welfare of Japan. Written informed consent was obtained from all participants. The protocol was approved by the main institutional review board (Nagoya University, no. 2014–0027) and by each participating facility. The study was registered with ClinicalTrials.gov (NCT02136563) and UMIN-CTR (UMIN000013464).

Patients with ND-CKD aged ≥20 years with an estimated glomerular filtration rate (eGFR) of <60 mL/min/1.73 m$^2$ (calculated using the Japanese equation [25]), who presented with renal anemia (Hb <11 g/dL), were enrolled from June 2014 to September 2016 and observed for 96 weeks after DA administration. The patients were excluded if they were scheduled for initiation of maintenance dialysis or to undergo kidney transplantation within 24 weeks after registration. Those with a history of ESA treatment and those undergoing treatment for malignant tumors, hematologic diseases, or hemorrhagic diseases were excluded. This study aimed to clarify the actual conditions of low ESA response cases in Japan, investigate factors relating to low ESA response cases, and search for new ERIs. Therefore, in consideration of securing data for each type of exploratory analysis, a sample size of 2,000 was set as the scale of research where evaluations of low ESA response cases were believed to be possible. The TREAT trial conducted internationally reported that renal events (end-stage renal failure or death) in the darbepoetin alfa group and the 2-year occurrence rate of CVD events was approximately 23–24% [16]. If the event occurrence rate was set as 13.1/100 people/year from this result, then the number of events observed in 2 years with a target registered number of cases of 2,000 would be 480–568 cases at a 95% confidence interval, which is believed to be a sufficient number of events for the planned analyses.

## DA administration

DA was administered according to the recommended regimen: 30 μg every 2 weeks for the initial dose; the dosage and duration should be adjusted thereafter to maintain Hb levels at ≥11 g/dL. The dose adjustment was left to the discretion of physicians in charge of each facility since this study was performed in a real-world clinical setting.

## Collection of data and events

Baseline patient characteristics were collected as previously described [23, 24]. Renal and CVD events were defined as follows: deterioration in renal function was defined as the initiation of maintenance dialysis, kidney transplantation, 50% decrease in eGFR, or eGFR of ≤6 mL/min/1.73 m$^2$. A fatal CVD event was described as death due to myocardial infarction, congestive heart failure, arrhythmia, cerebrovascular diseases, aortic dissection, other forms of cardiovascular diseases, ischemia in major organs, and sudden death. A nonfatal CVD event was determined as hospitalization due to myocardial infarction, angina pectoris, ischemic heart disease requiring invasive treatment, congestive heart failure, severe arrhythmia, atrial fibrillation, atrial flutter, aortic dissection, and ischemia of major organs.

## Candidate prognostic factors and equations for ESA hyporesponsiveness

We explored the prognostic factors for each event using the following variables: age, sex, presence of diabetes, arteriosclerosis, chronic inflammation, past history of malignancy, hemorrhagic lesion, collagen disease, rheumatoid arthritis, other myelosuppressive factors, immunosuppressive agent or steroid use, renin-angiotensin system inhibitor use, body weight, body mass index, serum albumin, Hb at 12 weeks, Hb change during 12 weeks, dose of DA at 12 weeks, total dose of DA over 12 weeks, logarithm-transformed eGFR, hs-CRP, NT-pro brain natriuretic peptide (BNP), folic acid, vitamin B12, iron, ferritin, transferrin saturation, and urinary protein-creatinine ratio. In this study, the responsiveness to ESA was calculated at 12 weeks, because the achieved hemoglobin level and the required dose of ESA in ND-CKD patients in daily clinical practice of Japan has been reported to become a steady state at about 3

months after starting ESA treatment [26].

$$4'. \text{ iEResI-2A} = \frac{\Delta Hb_{0-12}(g/dL) \times \text{body weight (kg)}}{\text{Total dose of DA during 12 weeks (μg)}}$$

$$5'. \text{ iEResI-2B} = \frac{\Delta Hb_{0-12}(g/dL)}{\text{Total dose of DA during 12 weeks (μg)}}$$

The following formulae were originally used to assess ESA hyporesponsiveness:

1. $\text{ERI-1A} = \frac{\text{Dose of DA at 12 weeks (μg)}}{\text{Concentration of Hb (g/dL) at 12 weeks} \times \text{body weight (kg)}}$

2. $\text{ERI-1B} = \frac{\text{Dose of DA at 12 weeks (μg)}}{\text{Concentration of Hb (g/dL) at 12 weeks}}$

3. $\Delta Hb\ 0-12\ (g/dL)$

4. $\text{ERI-2A} = \frac{\text{Total dose of DA during 12 weeks (μg)}}{\Delta Hb_{0-12}\ (g/dL) \times \text{body weight (kg)}}$

5. $\text{ERI-2B} = \frac{\text{Total dose of DA during 12 weeks (μg)}}{\Delta Hb_{0-12}\ (g/dL)}$

where $\Delta Hb\ 0-12\ (g/dL)$ = Hb (g/dL) at 12 weeks–Hb (g/dL) before DA administration. However, some patients showed decreased or no changes in the Hb levels ($\Delta Hb\ 0-12 < 0$ or $= 0$) during the 12-week DA administration; thus, these aforementioned formulae were not used in the data analysis. Instead, initial ESA response indices (iEResI) were defined as reciprocals of ERI-2A and ERI-2B.

In ERI-1A and ERI-1B, higher values correlate poorer responsiveness to ESA. In contrast, in $\Delta Hb\ 0-12$, iEResI-2A, and iEResI-2B, lower values indicate hyporesponsiveness.

## Statistical analysis

Baseline characteristics are presented as means ± standard deviations (SDs), medians (interquartile ranges), or numbers (percentages). The Cox proportional hazards model with covariates was applied to explore prognostic factors that influence the incidence of deterioration in renal function and CVD events. The models were examined using a stepwise method by adding and subtracting covariates as needed. These analyses were applied to investigate formulas to predict event incidence from the covariates.

The cut-off value for each ERI, iEResI, and linear predictor by prognostic factors in the Cox proportional hazards model increased gradually from minimum to maximum, to create time-dependent receiver operating characteristic (ROC) curves of survival time data until the occurrence of the events, and the areas under the curves (AUC) were compared. As the follow-up time was at least 2 years, AUC and optimum cut-off values were obtained from time-dependent ROC curves at 2 years. A patient with an ERI above or iEResI below the cut-off value was considered as having higher risk of renal and CVD event than those with lower ERI or higher iEResI. When the optimum cut-off value for ERI, iEResI, or linear predictor was determined, two groups were created based on this value ($<$ or $\geq$), cumulative survival curves were estimated using the Kaplan–Meier method, and yearly incidence rates and 95% confidence intervals (CIs) were estimated. In addition, a log-rank test was performed to compare the two groups; the hazard ratios (HRs) and 95% CIs were further estimated. These analyses were applied to investigate the predictive ability of each ERI, iEResI, and linear predictor at 2 years. All analyses were performed using SAS version 9.4 (SAS Institute, Cary, NC, USA), and P-values $<0.05$ were considered significant.

## Results

### Patients

Of the 1,980 patients enrolled in 168 facilities, 256 were excluded mainly due to the lack of data on Hb values at 0 and 12 weeks (84 ± 14 days). Finally, 1,724 patients were included in the analysis (Fig 1). The baseline characteristics are presented in Table 1.

In the mean age was 69.9 ± 12.0 years, and 58.8% of the participants were male. The etiologies of CKD were diabetic nephropathy (27.7%), chronic glomerulonephritis (23.2%), nephrosclerosis (23.5%), and polycystic kidney disease (5.6%). Past history of malignancy was reported in 12.3% of patients. Cardiovascular diseases were recorded as coronary artery disease, heart failure, stroke, and peripheral artery disease in 16.1%, 6.9%, 11.8%, and 10.9% of cases, respectively.

### Outcomes and risk factors

During the total study period, 139 patients died. Among them, 27 died due to lethal CVD events, and 112 cases with other causes of death were treated as censored cases. During the mean follow-up period of 96 weeks, renal and CVD events occurred in 683/1,724 (39.6%) and in 174/1,724 (10.1%) patients, respectively. The univariate and Cox proportional hazards model analyses for deterioration of renal function are presented in Table 2, and those for CVD events are shown in Table 3.

In the univariate analysis, several factors including physical characteristics, clinical data at 12 weeks of DA administration, past medical history, and renal function were significantly

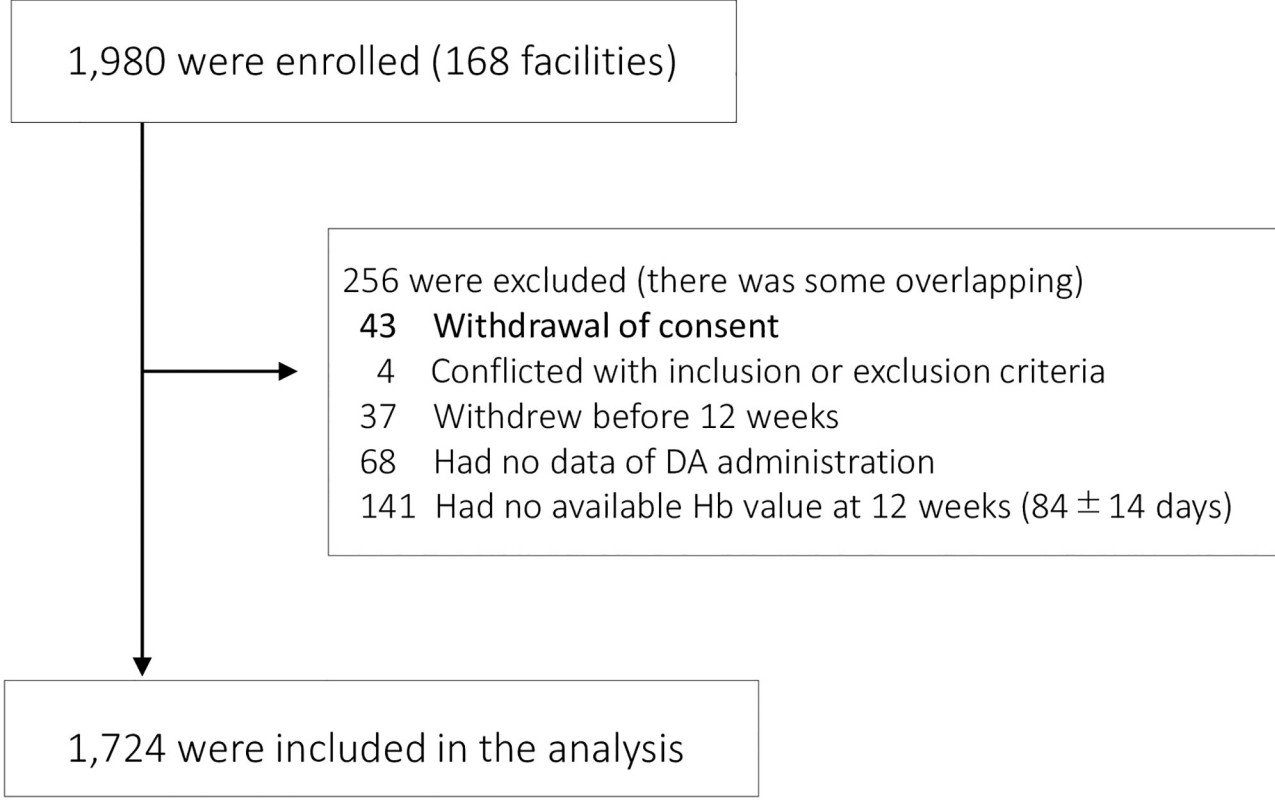

**Fig 1. Flow of participants.**

**Table 1.  Baseline characteristics.**

| | | Number of patients | Prevalence, mean, or median* |
|---|---|---|---|
| Age years | | 1724 | 69.9±12.0 |
| Male gender n, % | | - | 1013 (58.8) |
| Etiology of CKD | Diabetic nephropathy n, % | - | 477 (27.7) |
| | Chronic glomerulonephritis n, % | - | 400 (23.2) |
| | Nephrosclerosis n, % | - | 405 (23.5) |
| | Polycystic kidney disease n, % | - | 96 (5.6) |
| | Other n, % | - | 346 (20.1) |
| Smoking status | Current n, % | - | 188 (10.9) |
| | Ever n, % | - | 625 (36.3) |
| Diabetes | | - | 743 (43.1) |
| Malignancy (past history) | | - | 212 (12.3) |
| Cardiovascular disease | Coronary artery disease | - | 278 (16.1) |
| | Heart failure | - | 119 (6.9) |
| | Stroke | - | 204 (11.8) |
| | Peripheral artery disease | - | 188 (10.9) |
| RAS inhibitor use | Angiotensin II receptor blocker n, % | - | 983 (57.0) |
| | Angiotensin converting enzyme inhibitor n, % | - | 179 (10.4) |
| Hypoglycemic agent use | Dipeptidyl peptidase-4 inhibitor | - | 362 (21.0) |
| | Insulin | - | 196 (11.4) |
| Iron supplementation | | - | 250 (14.5) |
| Body mass index (kg/m$^2$) | | 1579 | 23.2±4.0 |
| Systolic arterial pressure (mmHg) | | 1626 | 134.4±19.1 |
| Diastolic arterial pressure (mmHg) | | 1624 | 71.3±12.4 |
| Creatinine (mg/dl) | | 1724 | 2.62 (1.88–3.63) |
| Estimated glomerular filtration rate (ml/min/1.73m$^2$) | | 1724 | 18.0 (17.8–25.3) |
| Hemoglobin (g/dl) | | 1724 | 9.8±0.9 |
| Albumin (g/dl) | | 1678 | 3.7±0.5 |
| Ferritin (ng/ml) | | 1672 | 96.5 (46.3–177.5) |
| Transferrin saturation (%) | | 1671 | 26.2 (20.6–31.9) |
| High sensitive C-reactive protein (ng/dl) | | 1673 | 575.0 (219.0–1790.0) |
| Folic acid (ng/ml) | | 1671 | 7.3 (5.6–10.1) |
| Vitamin B$_{12}$ (pg/ml) | | 1603 | 354.0 (259.0–499.0) |
| NT-proBNP* (pg/ml) | | 1673 | 517.0 (241.0–1160.0) |
| HbA1c (%) | | 1028 | 6.1±0.9 |
| Urinary protein-creatinine ratio (g/gCr) | | 1591 | 1.3 (0.4–3.0) |

associated with both subsequent renal and CVD events. However, in the multivariate analysis that adjusted for variables significantly associated with deterioration of renal function in univariate analysis, independent risk factors for this outcome were male sex, non-diabetes, past history of malignancy, eGFR, hemoglobin at 12 weeks, urinary protein–creatinine ratio, and total dose of DA in 12 weeks. In contrast, only male sex, arteriosclerosis, and NT-pro BNP remained independent risk factors for CVD events after adjusting for variables that were significantly associated in univariate analysis.

## Hyporesponsiveness to ESA and outcomes

Using each candidate formula for ESA hyporesponsiveness listed in the method, optimum cut-off values and AUCs for renal and CVD events were separately calculated by analyzing

**Table 2. Cox proportional hazards model analysis for deterioration of renal function.**

| Factor | Level or Unit | Univariate | | Multivariate* | |
|---|---|---|---|---|---|
| | | Hazard ratio (95%CI) | P value | Hazard ratio (95%CI) | P value |
| Male gender (reference, female) | Male | 2.040 (1.722–2.415) | p<0.001 | **2.507 (2.017–3.117)** | p<0.001 |
| Diabetes (reference, no) | Yes | 1.139 (0.976–1.331) | p = 0.099 | **0.646 (0.525–0.794)** | p<0.001 |
| Arteriosclerosis (reference, no) | Yes | 0.913 (0.774–1.078) | p = 0.282 | | |
| Chronic inflammation (reference, no) | Yes | 1.002 (0.837–1.199) | p = 0.984 | | |
| Past history of malignancy (reference, no) | Yes | 0.917 (0.722–1.165) | p = 0.479 | **1.368 (1.026–1.826)** | **p = 0.033** |
| Hemorrhagic lesion (reference, no) | Yes | 0.669 (0.334–1.343) | p = 0.259 | | |
| Collagen disease and rheumatoid arthritis (reference, no) | Yes | 0.416 (0.277–0.626) | p<0.001 | | |
| Other myelosuppressive factors (reference, no) | Yes | 0.557 (0.139–2.230) | p = 0.408 | | |
| Immunosuppressive agent or steroid use (reference, no) | Yes | 0.705 (0.549–0.905) | p = 0.006 | | |
| RAS inhibitor use (reference, no) | Yes | 1.101 (0.926–1.310) | p = 0.276 | | |
| Age | 10 years | 0.788 (0.742–0.837) | p<0.001 | | |
| Body weight | 10 kg | 1.239 (1.166–1.316) | p<0.001 | | |
| Body mass index | 1 kg/m² | 1.019 (0.999–1.039) | p = 0.062 | | |
| Log (Estimated glomerular filtration rate) | 1 mL/min/1.73m² | 0.073 (0.060–0.089) | p<0.001 | **0.087 (0.066–0.115)** | p<0.001 |
| Albumin | 1 g/dL | 0.452 (0.392–0.520) | p<0.001 | | |
| Hemoglobin at 12 weeks | 1 g/dL | 0.763 (0.712–0.819) | p<0.001 | **0.829 (0.753–0.912)** | p<0.001 |
| Hemoglobin change during 12 weeks | 1 g/dL | 0.914 (0.850–0.983) | p = 0.016 | | |
| Log (High-sensitivity C-reactive protein) | 1 ng/mL | 0.997 (0.944–1.053) | p = 0.917 | | |
| Log (NT-pro BNP) | 1 pg/mL | 1.467 (1.382–1.558) | p<0.001 | | |
| Log (Folic acid) | 1 ng/mL | 0.782 (0.664–0.921) | p = 0.003 | | |
| Log (Vitamin B12) | 1 pg/mL | 1.015 (0.865–1.190) | p = 0.859 | | |
| Log (Iron) | 1 µg/dL | 1.036 (0.828–1.295) | p = 0.759 | | |
| Log (Ferritin) | 1 ng/mL | 1.199 (1.094–1.315) | p<0.001 | | |
| Log (Transferrin saturation) | 1% | 1.539 (1.230–1.925) | p<0.001 | | |
| Log (Urinary protein-creatinine ratio) | 1 g/gCr | 2.096 (1.940–2.263) | p<0.001 | **2.062 (1.848–2.300)** | p<0.001 |
| Dose of DA at 12 weeks | 10 µg | 1.000 (0.979–1.021) | p = 1.000 | | |

(*Continued*)

**Table 2.** (Continued)

| Factor | Level or Unit | Univariate | | Multivariate* | |
|---|---|---|---|---|---|
| | | Hazard ratio (95%CI) | P value | Hazard ratio (95%CI) | P value |
| Total dose of DA during 12 weeks | 10 μg | 1.002 (0.994–1.010) | p = 0.610 | **1.017 (1.005–1.030)** | **p = 0.007** |

*: adjusted for variables significantly associated with deterioration of renal function in univariate analysis.

Abbreviations: RAS, renin-angiotensin system; NT-proBNP, amino-terminal pro-brain natriuretic peptide; DA, darbepoetin alfa

time-dependent ROC curves at 2 years. As shown in Table 4A and 4B, most of them were statistically significant in the survival analysis when patients were divided into two groups: those with the optimum cut-off value or more and those with less than that; none of their AUCs were considered extremely high.

Among these candidate formulae, ERI-1B, expressed by dividing the dose of DA by Hb level at the 12-week DA treatment, was the most statistically significant in predicting renal and CVD events. Prognosis of renal function of patients with ERI-1B of ≥5.1724 was significantly worse than those with lower ERI-1B (HR, 1.449; 95% CI, 1.231–1.705; P<0.0001). Similarly, patients with higher ERI-1B had worse prognosis for CVD events (HR, 1.719; 95% CI, 1.239–2.386; P = 0.0010). Moreover, the optimum cut-off values for deterioration of renal function (5.1724) and CVD events (5.2174) were close to each other (approximately 5.2). Conversely, the cut-off values for the two endpoints were obviously inconsistent in other candidate formulae compared to ERI-1B. The renal and CVD outcomes in patients with high and low ERI-1B values is presented in Fig 2A, 2B.

## Discussion

This prospective cohort study, based on real-world clinical data, defined the ESA hyporesponsive index in patients with ND-CKD, which may be clinically useful for the management of renal anemia. Although the study by Kilpatrick et al. has clearly reported that low ESA responsiveness in hemodialysis patients is a strong, independent predictor of mortality risk, it did not include ND-CKD patients [27]. In the prospective study by Minutolo et al., 194 patients with ND-CKD patients were operationally classified into 3 groups according to the ESA responsiveness and showed that the patients with the lowest tertile of responsiveness had poor renal prognosis [28]. Therefore, the present study is different from these previous studies regarding the subject and method to determine the low responders. As shown in Table 4A, the ratio of delta Hb/total dose of DA, iERes-2A and 2B in the present study, was significantly associated with renal outcome. This is consistent with those in the previous reports [27, 28]. However, the two values were not associated with CV events in this study. The study by Minutolo et al. did not investigate about CV events [28]. We do not have a concrete explanation for the difference between the study by Kilpatrick et al. and ours. This may be related to the very low number of CV events, just 10.1% in our study, whereas 34% of mortality and 66% cardiac-related hospitalization observed in the study by Kilpatrick et al [27]. A study with longer observation may be required to identify the risk factors for CV events in Japanese ND-CKD patients.

Among pre-defined candidate formulae for evaluation of ESA response, ERI-1B, simply dividing the DA dose by Hb at 12-week DA treatment, was significantly associated with poor renal and cardiovascular outcomes. Moreover, the cut-off values for both outcomes were close to each other only in this formula. Therefore, the definition is believed to be useful for

**Table 3. Cox proportional hazards model analysis for cardiovascular disease events.**

| Factor | Level or Unit | Univariate | | Multivariate* | |
|---|---|---|---|---|---|
| | | Hazard ratio (95%CI) | P value | Hazard ratio (95%CI) | P value |
| Male gender (reference, female) | Male | 1.825 (1.296–2.568) | p<0.001 | **1.648 (1.026–2.646)** | **p = 0.039** |
| Diabetes (reference, no) | Yes | 1.661 (1.213–2.274) | p = 0.002 | | |
| Arteriosclerosis (reference, no) | Yes | 2.340 (1.711–3.201) | p<0.001 | **2.561 (1.657–3.959)** | **p<0.001** |
| Chronic inflammation (reference, no) | Yes | 1.151 (0.808–1.639) | p = 0.435 | | |
| Past history of malignancy (reference, no) | Yes | 1.210 (0.777–1.883) | p = 0.399 | | |
| Hemorrhagic lesion (reference, no) | Yes | 0.360 (0.051–2.569) | p = 0.308 | | |
| Collagen disease and rheumatoid arthritis (reference, no) | Yes | 0.829 (0.437–1.573) | p = 0.565 | | |
| Other myelosuppressive factors (reference, no) | Yes | 0.000 (0.000—missing) | p = 0.978 | | |
| Immunosuppressive agent or steroid use (reference, no) | Yes | 0.802 (0.491–1.311) | p = 0.380 | | |
| RAS inhibitor use (reference, no) | Yes | 0.724 (0.522–1.004) | p = 0.053 | | |
| Age | 10 years | 1.386 (1.188–1.618) | p<0.001 | | |
| Body weight | 10 kg | 0.994 (0.865–1.142) | p = 0.930 | | |
| Body mass index | 1 kg/m$^2$ | 0.987 (0.945–1.030) | p = 0.545 | | |
| Log (Estimated glomerular filtration rate) | 1 mL/min/1.73m$^2$ | 0.545 (0.405–0.735) | p<0.001 | | |
| Albumin | 1 g/dL | 0.605 (0.456–0.805) | p<0.001 | | |
| Hemoglobin at 12 weeks | 1 g/dL | 0.746 (0.650–0.856) | p<0.001 | | |
| Hemoglobin change during 12 weeks | 1 g/dL | 0.877 (0.757–1.016) | p = 0.080 | | |
| Log (High-sensitivity C-reactive protein) | 1 ng/mL | 1.083 (0.971–1.208) | p = 0.154 | | |
| Log (NT-pro BNP) | 1 pg/mL | 1.873 (1.683–2.083) | p<0.001 | **1.843 (1.586–2.142)** | **p<0.001** |
| Log (Folic acid) | 1 ng/mL | 0.992 (0.725–1.357) | p = 0.959 | | |
| Log (Vitamin B12) | 1 pg/mL | 0.885 (0.632–1.238) | p = 0.474 | | |
| Log (Iron) | 1 µg/dL | 0.568 (0.376–0.858) | p = 0.007 | | |
| Log (Ferritin) | 1 ng/mL | 1.230 (1.017–1.487) | p = 0.033 | | |
| Log (Transferrin saturation) | 1% | 0.679 (0.451–1.023) | p = 0.064 | | |
| Log (Urinary protein-creatinine ratio) | 1 g/gCr | 1.253 (1.102–1.425) | p<0.001 | | |
| Dose of DA at 12 weeks | 10 µg | 1.027 (0.987–1.069) | p = 0.190 | | |

(*Continued*)

**Table 3.** (Continued)

| Factor | Level or Unit | Univariate | | Multivariate* | |
|---|---|---|---|---|---|
| | | Hazard ratio (95%CI) | P value | Hazard ratio (95%CI) | P value |
| Total dose of DA during 12 weeks | 10 μg | 0.995 (0.978–1.012) | p = 0.538 | | |

*: adjusted for variables significantly associated with deterioration of renal function in univariate analysis.

Abbreviations: RAS, renin-angiotensin system; NT-proBNP, amino-terminal pro-brain natriuretic peptide; DA, darbepoetin alfa

application in clinical practice. Other candidate formulae, such as accumulating or weight-adjusted DA doses, or those with change in the Hb level were not consistently significant for the association, and the cut-off values for both outcomes tended to be discrepant. We initially assumed that formulae using weight-adjusted DA dose was more correctly associated with the responsiveness to ESA than those with absolute ESA dose; however, this was incorrect. Although this study could not clearly provide the reason for this unexpected finding, this simple equation could estimate correctly, for example, apparent high responsiveness by weight loss due to undernutrition.

Accumulating evidence has indicated the negative prognostic value of a poor initial response to ESAs [29–33]. In the TREAT study, the definition of poor responsiveness to ESA in patients with ND-CKD was operationally defined as <2% in the first month after weight-based doses of DA, which was the lowest quartile of change in hemoglobin level. This result

**Table 4.** A. Optimum cut-off values for deterioration of renal function. B. Optimum cut-off values for cardiovascular disease events.

A.

| ERI, iEResI, or Linear predictor | Optimum cut-off value | HR (95% CI) | $p$ Log-rank test | AUC |
|---|---|---|---|---|
| ERI-1A | 0.0641 | 1.229 (1.090–1.548) | 0.0034 | 0.5311 |
| ERI-1B | 5.1724 | 1.449 (1.231–1.705) | <0.0001 | 0.5591 |
| Hb change during 12 weeks (g/dL) | 1.6 | 0.813 (0.687–0.961) | 0.0154 | 0.5371 |
| iEResI-2A | 0.776 | 0.738 (0.609–0.896) | 0.002 | 0.5354 |
| iEResI-2B | 0.0089 | 0.803 (0.688–0.938) | 0.0057 | 0.5439 |

B.

| ERI, iEResI, or Linear predictor | Optimum cut-off value | HR (95% CI) | $p$ Log-rank test | AUC |
|---|---|---|---|---|
| ERI-1A | 0.084 | 1.784 (1.242–2.561) | 0.0015 | 0.5813 |
| ERI-1B | 5.2174 | 1.719 (1.239–2.386) | 0.001 | 0.5694 |
| Hb change during 12 weeks (g/dL) | 2 | 0.711 (0.475–1.064) | 0.095 | 0.5339 |
| iEResI-2A | -1.3207 | | 0.3567 | 0.4379 |
| iEResI-2B | 0.0153 | 1.041 (0.712–1.521) | 0.0153 | 0.4652 |

Optimum cut-off values were calculated by time-dependent ROC curves at 2 years

A: Renal event

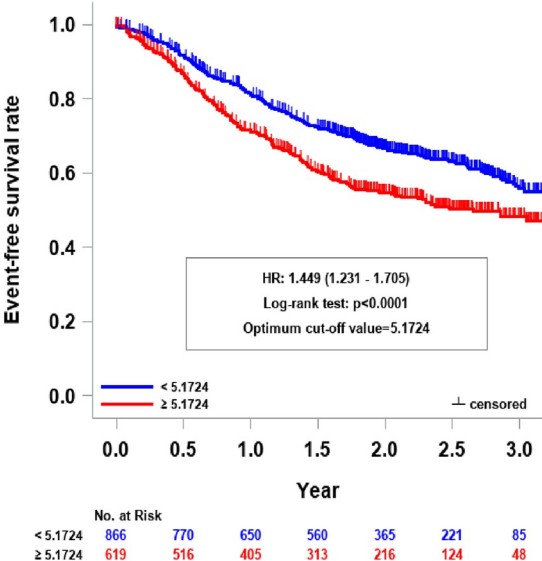

B: CVD event

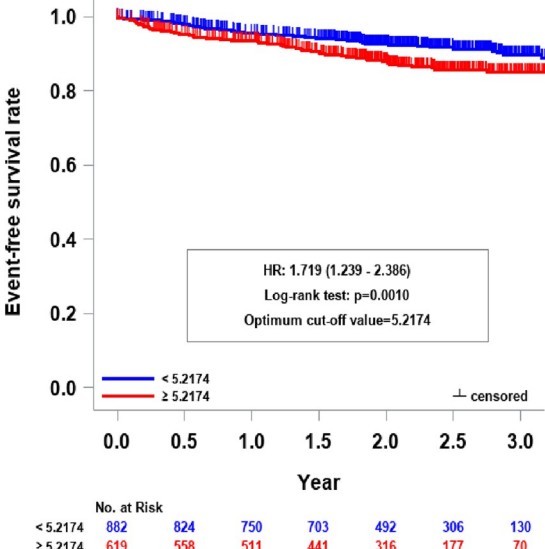

**Fig 2. Renal and CVD outcomes in patients with and without ESA hyporesponsiveness.** A, Renal survival was significantly worse in patients with high ESA hyporesponsive index (ERI-1B $\geq$5.1724) than in those with lower ERI-1B (log-rank test; P<0.0001). B, Event-free survival for CVD was significantly lower in patients with high ERI-1B ($\geq$5.2174) than in those with low ERI-1B (log-rank test; P = 0.0010). ESA, erythropoiesis-stimulating agents; CVD, cardiovascular disease.

clearly indicated that the poor initial response to DA was associated with an increased risk of death or CVD events as doses were escalated to meet the target hemoglobin levels. In contrast, multiple undeniable discrepancies between situations in TREAT and the real-world clinical practice in our country should be considered; for example, much higher doses of DA were administered (median dose, 232 and 167 µg for patients with poor and better responsiveness,

respectively), and the prevalence of a history of cardiovascular disease (>60% of patients) was higher in TREAT study. Considering that the 2-year incidence rate of CVD events in the DA group in the TREAT was 23–24%, and assuming that almost 2000 cases enrolled in our study would be observed for 2 years, we expected that CVD events would occur in 480–586 cases at a 95% CI. However, CVD events were observed in only 157 patients, whereas renal events occurred in 646 patients, implying that the patient characteristics and the actual state of medical care differ greatly. Based on these differences, we thought to directly apply the results of TREAT to our clinical practice. Thus, we attempted to explore an absolute definition of hyporesponsiveness to ESA, which is prognostically associated with the poor prognosis of renal and CVD outcomes, which to the best of our knowledge has never been provided.

The cut-off value of ERI-1B was 5.2, suggesting that a patient requiring ≥60 μg of DA to maintain a Hb level of 11 g/dL has a hyporesponsiveness to ESA, which is associated with a risk for progression to renal dysfunction and CVD events. A search for the causes of poor response and measures against them should be recommended in such patients. In addition to iron deficiency as the main cause of hyporesponsiveness to ESA, other factors that may be involved in the poor hematopoietic response should be considered, including the existence of malignancies, inflammatory diseases, bleeding, deficiencies of trace elements (i.e., vitamin B12, zinc, or copper), and hypoparathyroidism [34].

This study had several limitations. First, the dosage and interval of ESA administration and iron supplementation were not controlled, because this was designed as an observational cohort study, and the administration of ESA and iron supplementation were left to the discretion of physicians in charge. Thus, there is a possibility that some physicians titrate ESA and iron more rapidly than others, and this could confound the present observations. In fact, our previous study has shown that iron supplementation was an independent factor for better initial ESA responsiveness in this study population [24]. The main purpose of the present study was to define the hyporesponsiveness of ESA that associate with poor renal and CVD outcome in the real-world clinical setting of ND-CKD patients. The majority of physicians who participated in this study were nephrologists and proficient in treating anemia in patients with ND-CKD, following the guidelines of the Japanese Society of Dialysis and Transplantation 2015 [35], and the patient characteristics were representative of ND-CKD in our country [3, 36]. Therefore, we believe that the definition of hyporesponsiveness to ESA provided by this study is valid for application in daily clinical practice, at least in this country. Currently, we have yet to determine how the result of this study can be extrapolated to other countries. It has been shown that clinical practice of renal anemia treatment varied internationally, even within Western countries [37]. We assume the result of our study could be better extrapolated in countries where the clinical practice of ND-CKD does not differ from that in Japan, for example, ESA and iron are commonly used at low dose and mean level of eGFR at the introduction to dialysis is relatively low. Further investigation is required to examine the ethnic difference in the definition of ESA hyporesponsiveness. Second, this study could not clarify the reason why the independent risk factors identified by the Cox proportional hazard model analyses for the two endpoints, renal and CVD outcomes, were widely different from each other. In the univariate analyses, various factors were consistently associated with both outcomes; therefore, it is possible that the impact of each risk factor and the confounding relationships among them were different between the two outcomes. Third, as this method was based on direct estimation of ESA responsiveness, it was impossible to identify the "hyporesponders" before DA administration. Therefore, the baseline analysis of BRIGHTEN has investigated the contributing factors for initial responsiveness to ESA and reported that non-responders to DA accounted for 13.3% of patients with ND-CKD, whereas iron supplementation, low CRP, low NT-proBNP, and less proteinuria were predictive and modifiable factors associated with a

better initial response to DA [24]. The main purpose of this analysis was to establish a method for directly estimating the ESA hyporesponsiveness that is related to the poor prognosis of renal and CVD outcomes in patients with ND-CKD. Fourth, the area under the curve in Table 4A and 4B were not high, indicating that the predictive value of the ERI-1B may be limited. We aim to propose this index as one of independent prognostic factors useful when applied in combination with other well-known clinical risk factors.

In conclusion, hyporesponsiveness to ESA, which is associated with a risk for renal and CVD events in patients with ND-CKD, may be directly evaluated as the dose of DA divided by the Hb level at the 12-week DA treatment, with a cut-off value of 5.2. As ERI has been shown to be associated with multiple factors [24], a search for the modifiable causes of poor hematopoietic response and measures for collecting them should be considered in such patients.

## Supporting information

**S1 Checklist.**
(PDF)

**S1 Appendix.**
(PDF)

**S1 File.**
(PDF)

**S2 File.**
(PDF)

## Acknowledgments

We would like to express our deepest gratitude to the patients and investigators and staff at the study sites (listed below) for their contribution to the study.

Abashiri-Kosei General Hospital, Ageo Central General Hospital, Aichi Welfare Cooperative Agricultural Konan-Kosei Hospital, Akane Foundation Omachi Tsuchiya Clinic, Akebono Clinic, Anjo Kosei Hospital, Asahi University Hospital, Chibana Clinic, Chubu Rosai Hospital, Chuou Naika Clinic, Chutoen General Medical Center, Daido Clinic, Dokkyo Medical University Hospital, Dokkyo Medical University Saitama Medical Center, "Faculty of Medicine, University of Miyazaki Hospital", Fujinomiya City General Hospital, Fujita Health University Hospital, Fukui-ken Saiseikai Hospital, Fukuoka Red Cross Hospital, Fukuokahigashi Medical Center, Gunma University Hospital, Hamamatsu Medical Center, Hamamatsu University Hospital, Hamanomachi Hospital, Harasanshin Hospital, Heisei Hidaka clinic, Higashiosaka City Medical Center, Hiroshima General Hospital, Hiroshima Red Cross Hospital, Hiroshima University Hospital, Hokkaido Hospital, Hokkaido University Hospital, Hospital of the University of Occupational and Environmental Health, Hyogo Prefectural Nishinomiya Hospital, Ichiyokai Harada Hospital, Ikeda Hospital, "Ikeda Vascular Access, Dialysis and Internal Medicine Clinic", Iwate Medical University Hospital, Iwate Prefectural Central Hospital, JA Onomichi General Hospital, JA Toride Medical Center, Japanese Red Cross Ishinomaki Hospital, JCHO Chukyo Hospital, JCHO Osaka Hospital, JCHO Sendai Hospital, JCHO Tokuyama Central Hospital, Jichi Medical University Hospital, Joetsu General Hospital, JR Sapporo Hospital, Juntendo University Hospital, Kagawa Prefectual Central Hospital, Kagawa Rosai Hospital, Kagawa University Hospital, Kanazawa Medical University Hospital, Kanazawa Red Cross Hospital, Kanazawa University Hospital, Kansai Medical University Hospital, Kanto Rosai Hospital, Kasugai Municipal Hospital, Kawasaki Medical School Hospital, Kawasaki

Municipal Ida Hospital, Kawashima Hospital, Kitano Hospital, Kitasato Institute Medical Center Hospital, Kobe University Hospital, Koga Hospital 21, Komaki City Hospital, Koukan Clinic, Kure Kyosai Hospital, Kure Medical Center and Chugoku Cancer Center, Kurobe City Hospital, Kurume General Hospital, Kurume University Hospital, Kyorin University Hospital, Kyoto University Hospital, Kyushu University Hospital, Matsuyama Shimin Hospital, Mie University Hospital, Mizuho Hospital, Munakata Medical Hospital, Musashino Red Cross Hospital, Nagano Chuo Hospital, Nagano Red Cross Hospital, Nagaoka Chuo General Hospital, Nagaoka Red Cross Hospital, Nagasaki University Hospital, Nagoya City University Hospital, Nagoya Daini Red Cross Hospital, Nagoya Medical Center, Nagoya University Hospital, Nakadori General Hospital, Nanbugou kousei hospital, NHO Higashi-Hiroshima Medical Center, NHO Kanazawa Medical Center, NHO Kyoto Medical Center, NHO Tokyo Medical Center, Niigata Prefectural Sakamachi Hospital, Niigata Rinko Hospital, Niigata University Medical & Dental Hospital, Nippon Medical School Hospital, NTT Medical Center Tokyo, Nursing Medical Clinic Sakatsume-naika, Ogaki Municipal Hospital, Ohta Nishinouchi Hospital, Ojiya General Hospital, Okayama Medical Center, Okayama University Hospital, Okazaki City Hospital, Omuta City Hospital, Osaka City General Hospital, Osaka City University Hospital, Osaka General Medical Center, Osaka Medical College Hospital, Osaka Red Cross Hospital, Osaka University Hospital, Osaki Citizen Hospital, "Our Lady of Snow Social Medical Corporation, St. Mary Hospital", Rakuwakai Otowa Hospital, Saga University Hospital, Saga-Ken Medical Centre Koseikan, Saiseikai Nakatsu Hospital, Saiseikai Shimonoseki General Hospital, Saiseikai Yahata Hospital, Saiseikai Yokohamashi Nanbu Hospital, Saiseikai Yokohamashi Tobu Hospital, Saitama Medical University Hospital, Saitama-ken Saiseikai Kurihashi Hospital, Sapporo City General Hospital, Seirei Hamamatsu General Hospital, Sekishinnkai Kawasaki Saiwai Clinic, Sendai City Hospital, Sendai Red Cross Hospital, Shiga Universitu of Medical Science Hospital, Shimane University Hospital, Shonan Kamakura General Hospital, Showa University Fujigaoka Hospital, Showa University Hospital, Showa University Northern Yokohama Hospital, St. Luke's International Hospital, Tachikawa General Hospital, Teikyo University Hospital, The Jikei University Hospital, The St. Marianna University School of Medicine Hospital, The University of Tokyo Hospital, Tohoku University Hospital, Tokai University Hospital, Tokyo Medical University Hospital, Tokyo Medical University Ibaraki Medical Center, Tokyo Women's Medical University Hospital, Tonami General Hospital, Toranomon Hospital Kajigaya, Tosei General Hospital, Tottori University Hospital, Toyama City Hospital, Toyohashi Municipal Hospital, Toyota Kosei Hospital, Toyota Memorial Hospital, Tsushima Municipal Hospital, University of Tsukuba Hospital, Uonuma Kikan Hospital, Yamagata City Hospital Saiseikan, Yamagata University Hospital, Yamaguchi University Hospital, Yokkaichi Municipal Hospital, Yokohama City University Hospital, Yokohama City University Medical Center, Yokohama Municipal Citizen's Hospital, Yoshida Hospital.

We would like to thank Editage ([www.editage.com](www.editage.com)) for English language editing.

## Author Contributions

**Conceptualization:** Ichiei Narita, Terumasa Hayashi, Shoichi Maruyama, Takao Masaki, Masaomi Nangaku, Tomoya Nishino, Hiroshi Sato, Tadashi Sofue, Takashi Wada.

**Data curation:** Enyu Imai, Manabu Iwasaki, Kyoichi Mizuno, Hiroki Hase, Masahiro Kamouchi, Hiroyasu Yamamoto, Tatsuo Kagimura, Kenichiro Tanabe.

**Formal analysis:** Manabu Iwasaki, Hiroki Hase, Masahiro Kamouchi, Hiroyasu Yamamoto, Tatsuo Kagimura, Kenichiro Tanabe.

**Investigation:** Tomoya Nishino, Hiroshi Sato, Tadashi Sofue, Takashi Wada.

**Methodology:** Tatsuo Kagimura, Kenichiro Tanabe, Hideki Kato, Takehiko Wada, Tomoko Usui.

**Project administration:** Masaomi Nangaku, Hideki Kato, Takehiko Wada, Tomoko Usui.

**Supervision:** Tadao Akizawa, Hideki Hirakata, Yoshiharu Tsubakihara.

**Validation:** Shoichi Maruyama, Takao Masaki, Masaomi Nangaku, Tomoya Nishino, Tadashi Sofue, Kyoichi Mizuno, Hiroki Hase, Masahiro Kamouchi, Hiroyasu Yamamoto, Hideki Kato.

**Visualization:** Tatsuo Kagimura, Kenichiro Tanabe.

**Writing – original draft:** Ichiei Narita, Shoichi Maruyama, Masaomi Nangaku, Tomoya Nishino, Hiroshi Sato, Takashi Wada.

**Writing – review & editing:** Ichiei Narita, Terumasa Hayashi, Shoichi Maruyama, Takao Masaki, Masaomi Nangaku, Tomoya Nishino, Hiroshi Sato, Tadashi Sofue, Takashi Wada, Hideki Kato, Takehiko Wada, Tomoko Usui, Tadao Akizawa, Hideki Hirakata, Yoshiharu Tsubakihara.

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
