## [Decision Letter · Decision Letter 0]

10 May 2022

PONE-D-22-05315Hyporesponsiveness to erythropoiesis-stimulating agent in non-dialysis dependent CKD patients: The BRIGHTEN studyPLOS ONE

Dear Dr. Ichiei,

Thank you for submitting your manuscript to PLOS ONE. After careful consideration, we feel that it has merit but does not fully meet PLOS ONE’s publication criteria as it currently stands.  The referees found your work of potential interest but they also raised important critiques (see their reports below). Therefore, we invite you to submit a revised version of the manuscript that addresses clearly and unequivocally the points raised during the review process.

We look forward to receiving your revised manuscript.

Kind regards,

Gianpaolo Reboldi, MD, MSc, PhD

Academic Editor

PLOS ONE

Journal Requirements:

2. Thank you for stating the following in the Financial Disclosure:

(Translational Research Center for Medical Innovation has received a research grant, which was not specific for this study from Kyowa Kirin Co., Ltd. (KK). Kyowa Kirin was not involved in designing, data interpretation, and manuscript writing for this study.)  

We note that you received funding from a commercial source: (Kyowa Kirin)

Within this Competing Interests Statement, please confirm that this does not alter your adherence to all PLOS ONE policies on sharing data and materials by including the following statement: ""This does not alter our adherence to PLOS ONE policies on sharing data and materials.” (as detailed online in our guide for authors http://journals.plos.org/plosone/s/competing-interests).  If there are restrictions on sharing of data and/or materials, please state these. Please note that we cannot proceed with consideration of your article until this information has been declared. 

(Dr Narita reported receiving lecture fee and grant from Kyowa Kirin (KK). Dr Hayashi reported receiving honoraria from KK. Dr Maruyama received honoraria and subsidies or donation from KK. Dr Masaki reported receiving lecture fee and grant from KK. Dr Nangaku reported receiving honoraria, manuscript fees, and subsidies or donations from KK. Dr Nishino reported receiving lecture fee and grant from KK. Dr Sofue reported receiving grant from KK. Dr Takashi Wada reported receiving honoraria and subsidies or donation from KK. Dr Hase reported receiving honoraria from KK. Dr Yamamoto reported receiving honoraria and manuscript fees from KK. Dr Takehiko Wada reported receiving honoraria from KK. Dr Akizawa received honoraria and manuscript fees from KK. Dr Tsubakihara received honoraria from KK.)

5. Please include your tables as part of your main manuscript and remove the individual files. Please note that supplementary tables (should remain/ be uploaded) as separate "supporting information" files".

Reviewers' comments:

Reviewer's Responses to Questions

**Comments to the Author**

1. Is the manuscript technically sound, and do the data support the conclusions?

Reviewer #1: Partly

Reviewer #2: Partly

Reviewer #3: Yes

Reviewer #4: Partly

2. Has the statistical analysis been performed appropriately and rigorously? 

Reviewer #1: No

Reviewer #2: No

Reviewer #3: Yes

Reviewer #4: I Don't Know

3. Have the authors made all data underlying the findings in their manuscript fully available?

Reviewer #1: No

Reviewer #2: No

Reviewer #3: Yes

Reviewer #4: Yes

4. Is the manuscript presented in an intelligible fashion and written in standard English?

Reviewer #1: No

Reviewer #2: Yes

Reviewer #3: Yes

Reviewer #4: Yes

5. Review Comments to the Author

Reviewer #1: My comments are as follows:

1. The manuscript considers data analysis from a prospective observational study. More justification needed why it is submitted as a Clinical Trial. There doesn't appear to be any randomization.

2. It was a bit strange to see a manuscipt without any sample size/power statement, based on a targeted effect size that the authors wanted to achieve. Sample size should consider the primary outcome of interest.

3. Cox PH model was fitted; I do not see any assessments of the proportionality assumptions. Such assessments (repoted p-values) are easily available in any standard software (SAS, R, etc).

4. Looks like the study is multi-center, which implies that a frailty Cox PH model fitting may also be appropriate. Why was a frailty model not considered (one that factors in the cluster effect)? If the authors don't like to fit it, plausible justifications are necessary.

5. In the results section, any statement on HRs should be followed by the corresponding 95% CI, and the p-value. Please double check.

Reviewer #2: Major criticisms:

1) Regarding the timing for the evaluation of hyporesponsiveness to ESA, in the mentioned TREAT study, the definition of poor initial response to ESA was established at the first month after two doses of ESA and similarly in a study by Minutolo et al (NDT 27: 2880-2886, 2012) in non-dialysis patients “the predictive value of ESA-R calculated at first control visit is comparable with that measured after the 6-month observation period…” These findings are clinically relevant as it allows to obtain the same prognostic information some months earlier; therefore, it is necessary to evaluate the timing for the evaluation of ESA hyporesponsiveness index at first control visit and not only at 12 weeks.

2) The Authors state that their study defined the ESA hyporesponsiveness index “for the first time”. This sentence is uncorrected. The mentioned prospective study by Minutolo et al in fact evaluated the association between Hb change/time/ ESA dose (so called ESA-R) and the time to ESRD as primary outcome; moreover, also the study by Kilpatrik R et al from the NHC trial (cJASN 3: 1077-1083, 2008) evaluated the association between ratio of Hct change per ESA dose increase and mortality as primary outcome in dialysis patients.

3) It is surprising that -in the persent study- the ratio of delta Hb/ total dose of DA is not associated with the renal and cardiovascular outcomes in contrast with the mentioned previous findings from the literature (Minutolo R et al and Kilpatrik R). What is the explanation for these relevant differences respect to the results of literature?

4) Is it possible to estimate, for example, the increase of the risk for progression to renal failure and, similarly, CVD events in a patient with a normal cut-off value of 4.0 respect to a patient with an altered cut-off value of 6.0? This information is important since the statistical significance of the results does not necessarily imply its relevance for application in clinical practice.

5) The Figures 2A and 2B show a greater event free survival for renal event and for CVD event, respectively, in patients with ratio Hb/DA dose greater or equal 5.1724 in red! It is the opposite of the results reported in the present study.

6) In Figure 2A, the probability of event free survival for renal event is overlapping between patients with ratio Hb/DA dose lower or greater/equal 5.1724 in the range of 3-4 years of follow-up. What is the explanation of this phenomenon?

7) Instead of Kaplan-Meyer Figure 2A and 2B, not adjusted for confounding variables, it is more useful to represent the relative splines of risk, adjusted for variables.

8) A limitation of this study is due to the lack of data of secondary hyperparathyroidism. This complication is a potential cause of reduced responsiveness to ESA, as shown also in the conservative stage of chronic kidney disease (Di Iorio BR et al: Kidney Int 64:1822-1828, 2003).

9) In the Discussion (lines 252-253), the Authors state that “… this simple equation could compensate the variation in body mass and the influence of nutritional study.” It is not clear the significance of this sentence and, therefore, it is necessary to explain this point.

Reviewer #3: The paper by Narita Ichiei et al summarizes the results of BRIGHTEN Study, a multicenter, prospective observational study aimed at establishing a pertinent definition for hyporesponsiveness to ESAs that accurately predicts poor renal outcome and CVD events in patients with ND-CKD in a real-world clinical setting in Japan. Several candidate formulae for ESA responsiveness have been tested and the most statistically significant in predicting renal and CVD events was applied to the study population . With the multivariate analysis the authors tried to define the main predictors of renal and cardiac endpoints. It is striking that while the predictors of renal end points were clearly identifiable, the predictors of cardiac events were only male and NT-pro BNP.. This is related to the very low number of CV events which partially biased the study results. The formula identified as the most predictive of ESA hyporesponsiveness is not particularly different from those most commonly used. The study appears rigorous and well conducted even if not of particular innovative value. However, the standard of the work is good and confirms the value of monitoring ESA hyporesponsiveness as a predictor of events in ND-CKD patients.

Reviewer #4: In this paper, the Authors evaluated hyporesponsiveness to ESA in patients with ND-CKD by applying a new method. Darbepoetin alfa (DA) and iron supplement administrations were provided according to the recommendation of the attached document and the guidelines of JSDT (Japanese Society of Dialysis and Transplantation). The primary outcomes were progression of renal dysfunction and major adverse cardiovascular events. ESA responsiveness was assessed using pre-defined candidate formulae. Formula defining hyporesponsiveness obtained by dividing the dose of DA by hemoglobin (Hb) level at the 12-week DA treatment proved to be statistically significant in predicting renal and CVD events, with cut-off values for both events close to 5.2.

The work is of potential interest but requires some revisions.

The main drawback of the study is that the dosage and interval of both ESA administration and iron supplementation were not controlled and left to the discretion of physicians. Dosage of DA at 12-week is of critical value for evaluation of ESA responsiveness during this study and might have been influenced by different attitudes of physicians to titrate DA and iron. Furthermore, there is no mention on iron status and on the relationship between iron supplementation and response to DA and its effect on DA dosage.

The Authors report at line 222 that “Prognosis of renal function of patients with ERI-1B of ≥5.1724 was significantly worse than those with lower ERI-1B” and that (line 225) “Similarly, patients with higher ERI-1B had worse prognosis for CVD events”. However, examining Figure 2 A and 2B it would appear the opposite. A comment is appropriate.

Likewise, the meaning and the relevance of optimum cut-off values should be better explained.

How was response to DA evaluated?

It seems that some patients were not naïve to ESA treatment. If so, how was their response to previous ESA? And if their previous response was good, why to switch them?

Lines 102-104: “The patients were scheduled for initiation of maintenance dialysis or to undergo kidney transplantation within 24 weeks after registration.“ This is not clear.

How can the results of the study be extrapolated to Western countries?

6. PLOS authors have the option to publish the peer review history of their article (what does this mean?). If published, this will include your full peer review and any attached files.

Reviewer #1: No

Reviewer #2: No

Reviewer #3: No

Reviewer #4: No

---

## [Author Response · Author response to Decision Letter 0]

18 Jul 2022

RESPONSES TO THE COMMENTS 

The original comments of the reviewers and our responses to the comments are as follows:

Responses to Reviewer # 1’s Comments:

1. The manuscript considers data analysis from a prospective observational study. More justification needed why it is submitted as a Clinical Trial. There doesn't appear to be any randomization.

Response: We appreciate your comments. As stated in the abstract, this was an observational study, and therefore, no randomization was performed.

2. It was a bit strange to see a manuscipt without any sample size/power statement, based on a targeted effect size that the authors wanted to achieve. Sample size should consider the primary outcome of interest.

Response: We appreciate your comments. Since this is not a confirmatory study, we did not design the sample size by power or effect size. However, we have included the following statement in the protocol and added these sentences to the Materials and methods section in the revised version (lines 106–117).

“This study aimed to clarify the actual conditions of low ESA response cases in Japan, investigate factors relating to low ESA response cases, and search for new ERIs. Therefore, in consideration of securing data for each type of exploratory analysis, a sample size of 2,000 was set as the scale of research where evaluations of low ESA response cases were believed to be possible. The TREAT trial conducted internationally reported that renal events (end-stage renal failure or death) in the darbepoetin alfa group and the 2-year occurrence rate of CVD events was approximately 23–24% [16]. If the event occurrence rate was set as 13.1/100 people/year from this result, then the number of events observed in 2 years with a target registered number of cases of 2,000 would be 480–568 cases at a 95% confidence interval, which is believed to be a sufficient number of events for the planned analyses.”

3. Cox PH model was fitted; I do not see any assessments of the proportionality assumptions. Such assessments (repoted p-values) are easily available in any standard software (SAS, R, etc).

Response: We appreciate your comments. Here, we show the log(-log(survival)) plot and the Schoenfeld residuals plot for both renal function and CVD events as follows. The X-axis of log(-log(survival)) plot is a logarithmic scale; hence, the early period of the survival time is displayed longer (only the axis is a logarithmic scale, and the displayed values are not logarithmically converted). We did not think there was a problem with the assumption of proportional hazards. The p-value of the chi-square test for renal function was <0.05 (p = 0.046), which was considered to be due to the large sample size (n=1,687).

4. Looks like the study is multi-center, which implies that a frailty Cox PH model fitting may also be appropriate. Why was a frailty model not considered (one that factors in the cluster effect)? If the authors don't like to fit it, plausible justifications are necessary.

Response: We appreciate your comment. We did not analyze the frailty Cox PH model with a site as a cluster variable because it was not included in the analysis plan; however, we have conducted it in response to your suggestion.

As a result, the hazard ratio and its 95% confidence interval for renal function were 1.516 (1.275–1.803) and for CVD events were 1.807 (1.287–2.535), and the P values were <0.0001 and 0.0006, respectively. Since the results of our manuscript and those of the frailty Cox PH model seemed to be not considerably different, we would like to use the results of our manuscript corresponding to the analysis plan.

Another reason for hesitation in presenting the results of the frailty model was the presence of the sites which have very small number of patients. The study enrolled patients at 168 sites, but fewer than 4 patients at 48 sites.

5. In the results section, any statement on HRs should be followed by the corresponding 95% CI, and the p-value. Please double check.

Response: Thank you for your valuable comment. We have added 95% confidence intervals for hazard ratios to Tables 2 and 3 (pages 13, 14).

 

Responses to Reviewer # 2’s Comments:

1) Regarding the timing for the evaluation of hyporesponsiveness to ESA, in the mentioned TREAT study, the definition of poor initial response to ESA was established at the first month after two doses of ESA and similarly in a study by Minutolo et al (NDT 27: 2880-2886, 2012) in non-dialysis patients “the predictive value of ESA-R calculated at first control visit is comparable with that measured after the 6-month observation period…” These findings are clinically relevant as it allows to obtain the same prognostic information some months earlier; therefore, it is necessary to evaluate the timing for the evaluation of ESA hyporesponsiveness index at first control visit and not only at 12 weeks.

Response: Thank you for pointing out this significant issue. As you have pointed out, it would be better if the ESA-R could be determined at an earlier timing. However, in the present study, we evaluated the responsiveness to ESA at 12 weeks, because the achieved hemoglobin level and the required dose of ESA in ND-CKD patients in daily clinical practice of Japan has been reported to become a steady state at about 3 months after starting ESA treatment（Akizawa T, et al. 2011）. We have added this point in the Materials and methods section in the revised version (page 8, lines 148–151 and added a relevant paper (Akizawa T, et al. : Ther Apher Dial. 2011; 15: 431-440 PMID:21974695) to the reference list.

“In this study, ｔhe responsiveness to ESA was calculated at 12 weeks, because the achieved hemoglobin level and the required dose of ESA in ND-CKD patients in daily clinical practice of Japan has been reported to become a steady state at about 3 months after starting ESA treatment [26].”

2) The Authors state that their study defined the ESA hyporesponsiveness index “for the first time”. This sentence is uncorrected. The mentioned prospective study by Minutolo et al in fact evaluated the association between Hb change/time/ ESA dose (so called ESA-R) and the time to ESRD as primary outcome; moreover, also the study by Kilpatrik R et al from the NHC trial (cJASN 3: 1077-1083, 2008) evaluated the association between ratio of Hct change per ESA dose increase and mortality as primary outcome in dialysis patients.

Response: Thank you for your thoughtful comment. According to the comment, we have added a discussion on the relevant points as shown below (lines 257–264, and we deleted the phrase “for the first time” and modified the references accordingly.

“Although the study by Kilpatrick et al. has clearly reported that low ESA responsiveness in hemodialysis patients is a strong, independent predictor of mortality risk, it did not include ND-CKD patients [27]. In the prospective study by Minutolo et al., 194 patients with ND-CKD patients were operationally classified into 3 groups according to the ESA responsiveness and showed that the patients with the lowest tertile of responsiveness had poor renal prognosis [28]. Therefore, the present study is different from these previous studies regarding the subject and method to determine the low responders.”

3) It is surprising that -in the persent study- the ratio of delta Hb/ total dose of DA is not associated with the renal and cardiovascular outcomes in contrast with the mentioned previous findings from the literature (Minutolo R et al and Kilpatrik R). What is the explanation for these relevant differences respect to the results of literature?

Response: Thank you for your valuable comment. We have added the discussions on this point to the revised version as below (lines 264–273). 

“As shown in Table 4A, the ratio of delta Hb/total dose of DA, iERes-2A and 2B in the present study, was significantly associated with renal outcome. This is consistent with those in the previous reports [27; 28]. However, the two values were not associated with CV events in this study. The study by Minutolo et al. did not investigate about CV events [28]. We do not have a concrete explanation for the difference between the study by Kilpatrick et al. and ours. This may be related to the very low number of CV events, just 10.1% in our study, whereas 34% of mortality and 66% cardiac-related hospitalization observed in the study by Kilpatrick et al [27]. The study with longer observation may be required to identify the risk factors for CV events in Japanese ND-CKD patients.” 

4) Is it possible to estimate, for example, the increase of the risk for progression to renal failure and, similarly, CVD events in a patient with a normal cut-off value of 4.0 respect to a patient with an altered cut-off value of 6.0? This information is important since the statistical significance of the results does not necessarily imply its relevance for application in clinical practice.

Response: Thank you for your critical comment. In renal function, the hazard ratio and its 95% confidence interval for a cutoff value of 4 was 1.397 (1.187–1.646), and the hazard ratio and its 95% confidence interval for a cutoff value of 6 was 1.224 (1.013–1.479).

For CVD events, the hazard ratio and its 95% confidence interval for a cutoff value of 4 was 1.453 (1.044–2.023), and the hazard ratio and its 95% confidence interval for a cutoff value of 6 was 1.574 (1.104–2.245).

5) The Figures 2A and 2B show a greater event free survival for renal event and for CVD event, respectively, in patients with ratio Hb/DA dose greater or equal 5.1724 in red! It is the opposite of the results reported in the present study.

Response: We appreciate your comment. As you have pointed out, there was an error in the group information; thus, we have corrected it. 

 

6) In Figure 2A, the probability of event free survival for renal event is overlapping between patients with ratio Hb/DA dose lower or greater/equal 5.1724 in the range of 3-4 years of follow-up. What is the explanation of this phenomenon?

Response: We appreciate the reviewer’s comment on this point. The follow-up period in this study was set at 2 years, and for most cases, it was about 2 years. For this reason, the number of cases at risk after 3 years is very small for statistical analysis. Therefore, in the graph of renal function, the overlapping phenomenon after 3 years is not reliable. To avoid misunderstanding by readers, we would like to limit the X-axis of the graphs to 3 years as indicated below.

7) Instead of Kaplan-Meyer Figure 2A and 2B, not adjusted for confounding variables, it is more useful to represent the relative splines of risk, adjusted for variables.

Response: We wish to thank the reviewer for this comment. The factors related to the hyporesponsiveness to ESA were described in our previous paper (Hayashi T, et al. Clin Exp Nephrol 2021), and with that in mind, the main objective of this paper was to determine which ERI formula is best and which cutoff value. Hyporesponsiveness is a phenomenon caused by a variety of factors, including complications, and adjusting for factors that cause hyporesponsiveness may miss the relevance of the hyporesponsiveness index and the event. Factors that are independent of hyporesponsiveness and associated with the event could be confounding factors and should be considered in the analysis; however, we have not identified them at this time.

We drew splines (restricted cubic splines) showing the relationship between ERI-1B and risk (hazard ratio). Using an information criterion AICc, the lowest order model that could be computed was selected for both renal function and CVD events (see figure below).

We used the Youden Index of the ROC curve to determine the optimal cut-off value. Our choice of 5.1 or 5.2 is slightly below the peak risk for both renal function and CVD events in estimated spline curves. Patients with values greater than the optimal cut-off value can be considered high-risk patients, which appears to be an outcome that supports our suggested cut-off value.

8) A limitation of this study is due to the lack of data of secondary hyperparathyroidism. This complication is a potential cause of reduced responsiveness to ESA, as shown also in the conservative stage of chronic kidney disease (Di Iorio BR et al: Kidney Int 64:1822-1828, 2003).

Response: Thank you for your valuable comment. We have added “hypoparathyroidism” as a potential cause of hyporesponsiveness to ESA in the discussion (line 316) and changed the reference list accordingly.

“and hypoparathyroidism [34].”

9) In the Discussion (lines 252-253), the Authors state that “… this simple equation could compensate the variation in body mass and the influence of nutritional status...” It is not clear the significance of this sentence and, therefore, it is necessary to explain this point.

Response: My apologies for the confusion. We have modified the description on this point to make it clearer, as described below (lines 281–286). 

“We initially assumed that formulae using weight-adjusted DA dose was more correctly associated with the responsiveness to ESA than those with absolute ESA dose; however, this was incorrect. Although this study could not clearly provide the reason for this unexpected finding, this simple equation could estimate correctly, for example, apparent high responsiveness by weight loss due to undernutrition.”

Responses to Reviewer # 3’s Comments:

The paper by Narita Ichiei et al summarizes the results of BRIGHTEN Study, a multicenter, prospective observational study aimed at establishing a pertinent definition for hyporesponsiveness to ESAs that accurately predicts poor renal outcome and CVD events in patients with ND-CKD in a real-world clinical setting in Japan. Several candidate formulae for ESA responsiveness have been tested and the most statistically significant in predicting renal and CVD events was applied to the study population . With the multivariate analysis the authors tried to define the main predictors of renal and cardiac endpoints. It is striking that while the predictors of renal end points were clearly identifiable, the predictors of cardiac events were only male and NT-pro BNP.. This is related to the very low number of CV events which partially biased the study results. The formula identified as the most predictive of ESA hyporesponsiveness is not particularly different from those most commonly used. The study appears rigorous and well conducted even if not of particular innovative value. However, the standard of the work is good and confirms the value of monitoring ESA hyporesponsiveness as a predictor of events in ND-CKD patients.

Response: Thank you for your precise review and useful comments. We agree with Reviewer 3 that the possible reason why the predictors of CV events identified in the present study was only male and NT-pro BNP was the low number of CV events in this population. A study with longer observation time may be warranted to identify the risk factors for CV events in Japanese ND-CKD patients. We have added this point to the discussion in the revised manuscript (lines 272–273).

“A study with longer observation may be required to identify the risk factors for CV events in Japanese ND-CKD patients.”

Responses to Reviewer # 4’s Comments:

In this paper, the Authors evaluated hyporesponsiveness to ESA in patients with ND-CKD by applying a new method. Darbepoetin alfa (DA) and iron supplement administrations were provided according to the recommendation of the attached document and the guidelines of JSDT (Japanese Society of Dialysis and Transplantation). The primary outcomes were progression of renal dysfunction and major adverse cardiovascular events. ESA responsiveness was assessed using pre-defined candidate formulae. Formula defining hyporesponsiveness obtained by dividing the dose of DA by hemoglobin (Hb) level at the 12-week DA treatment proved to be statistically significant in predicting renal and CVD events, with cut-off values for both events close to 5.2.

The work is of potential interest but requires some revisions.

1. The main drawback of the study is that the dosage and interval of both ESA administration and iron supplementation were not controlled and left to the discretion of physicians. Dosage of DA at 12-week is of critical value for evaluation of ESA responsiveness during this study and might have been influenced by different attitudes of physicians to titrate DA and iron. Furthermore, there is no mention on iron status and on the relationship between iron supplementation and response to DA and its effect on DA dosage.

Response: Thank you for your comment. We agree with the comment that iron status is a significant factor for ESA responsiveness. In fact, our previous study has shown that iron supplementation was an independent factor for better initial ESA responsiveness in this study population (Hayashi T, et al. Clin Exp Nephrol 2021). According to the comments, we have added a description in the discussion (lines 322–326) and deleted the term “However” in the relative paragraph. 

“In fact, our previous study has shown that iron supplementation was an independent factor for better initial ESA responsiveness in this study population [24]. The main purpose of the present study was to define the hyporesponsiveness of ESA that associate with poor renal and CVD outcome in the real-world clinical setting of ND-CKD patients.”

2. The Authors report at line 222 that “Prognosis of renal function of patients with ERI-1B of ≥5.1724 was significantly worse than those with lower ERI-1B” and that (line 225) “Similarly, patients with higher ERI-1B had worse prognosis for CVD events”. However, examining Figure 2 A and 2B it would appear the opposite. A comment is appropriate.

Response: Thank you for pointing this serious error. We have corrected Figure 2A and 2B.

3. Likewise, the meaning and the relevance of optimum cut-off values should be better explained.

Response: Thank you for your valuable comment. We have added descriptions on the relevance of optimum cut-off value in the Materials and methods section (lines 177–179). 

“A patient with an ERI above or iEResI below the cut-off value was considered as having higher risk of renal and CVD event than those with lower ERI or higher iEResI.” 

4. How was response to DA evaluated?

Response: We thank you for the comment. We have added this point in the Materials and methods section of the revised version as indicated below (lines 148–151) and modified the reference list accordingly.

“In this study, the responsiveness to ESA was calculated at 12 weeks, because the achieved hemoglobin level and the required dose of ESA in ND-CKD patients in daily clinical practice of Japan has been reported to become a steady state at about 3 months after starting ESA treatment [26].”

5. It seems that some patients were not naïve to ESA treatment. If so, how was their response to previous ESA? And if their previous response was good, why to switch them?

Response: In this study, patients with a history of ESA treatment prior to the study were excluded as described in the Materials and methods section of the original version (lines 104–106).

6. Lines 102-104: “The patients were scheduled for initiation of maintenance dialysis or to undergo kidney transplantation within 24 weeks after registration.“ This is not clear.

Response: My apologies for the unclear statement. We have revised the sentence as follows (lines 102–104).

“The patients were excluded if they were scheduled for initiation of maintenance dialysis or to undergo kidney transplantation within 24 weeks after registration.”

7. How can the results of the study be extrapolated to Western countries?

Response: Thank you for your insightful comment. Currently, we do not have the definite answer to this question. It has been shown that clinical practice of renal anemia treatment widely varied, even within Western countries {Wong, 2020 #856}. We have added this point in the discussion as indicated below (lines 332–339) and modified the reference list accordingly.

“Currently, we have yet to determine how the result of this study can be extrapolated to other countries. It has been shown that clinical practice of renal anemia treatment varied internationally, even within Western countries [37]. We assume the result of our study could be better extrapolated in countries where the clinical practice of ND-CKD does not differ from that in Japan, for example, ESA and iron are commonly used at low dose and mean level of eGFR at the introduction to dialysis is relatively low. Further investigation is required to examine the ethnic difference in the definition of ESA hyporesponsiveness.”

---

## [Decision Letter · Decision Letter 1]

5 Aug 2022

PONE-D-22-05315R1Hyporesponsiveness to erythropoiesis-stimulating agent in non-dialysis dependent CKD patients: The BRIGHTEN studyPLOS ONE

Dear Dr. Ichiei,

Thank you for submitting your manuscript to PLOS ONE. After careful consideration, we feel that it has merit but does not fully meet PLOS ONE’s publication criteria as it currently stands. I do appreciate your revision and replies to all previous comments and remarks, but there is one remaining issue raised by one of the referees, who was not fully convinced by your revision. I'd suggest a quick reply to the referee's question "I still don't understand why the manuscript has been submitted as a "Clinical Trial", versus a standard Research Paper". 

We look forward to receiving your revised manuscript.

Kind regards,

Gianpaolo Reboldi, MD, MSc, PhD

Academic Editor

PLOS ONE

Journal Requirements:

Reviewers' comments:

Reviewer's Responses to Questions

**Comments to the Author**

1. If the authors have adequately addressed your comments raised in a previous round of review and you feel that this manuscript is now acceptable for publication, you may indicate that here to bypass the “Comments to the Author” section, enter your conflict of interest statement in the “Confidential to Editor” section, and submit your "Accept" recommendation.

Reviewer #1: (No Response)

Reviewer #4: All comments have been addressed

2. Is the manuscript technically sound, and do the data support the conclusions?

Reviewer #1: (No Response)

Reviewer #4: (No Response)

3. Has the statistical analysis been performed appropriately and rigorously? 

Reviewer #1: (No Response)

Reviewer #4: (No Response)

4. Have the authors made all data underlying the findings in their manuscript fully available?

Reviewer #1: (No Response)

Reviewer #4: (No Response)

5. Is the manuscript presented in an intelligible fashion and written in standard English?

Reviewer #1: (No Response)

Reviewer #4: (No Response)

6. Review Comments to the Author

Reviewer #1: The authors were somewhat responsive to my previous comments. I still don't understand why the manuscript has been submitted as a "Clinical Trial", versus a standard Research Paper. Please clarify further.

Reviewer #4: (No Response)

7. PLOS authors have the option to publish the peer review history of their article (what does this mean?). If published, this will include your full peer review and any attached files.

Reviewer #1: No

Reviewer #4: No

---

## [Author Response · Author response to Decision Letter 1]

12 Aug 2022

RESPONSES TO THE COMMENTS 

The original comments of the reviewers and our responses to the comments are as follows:

Responses to Reviewer # 1’s Comments:

#1: The authors were somewhat responsive to my previous comments. I still don't understand why the manuscript has been submitted as a "Clinical Trial", versus a standard Research Paper. Please clarify further.

Response: We appreciate for your comments and agree with your criticism. We would like to change the category.

---

## [Editor Report · Decision Letter 2]

8 Nov 2022

Hyporesponsiveness to erythropoiesis-stimulating agent in non-dialysis dependent CKD patients: The BRIGHTEN study

PONE-D-22-05315R2

Dear Dr. Ichiei,

We’re pleased to inform you that your manuscript has been judged scientifically suitable for publication and will be formally accepted for publication once it meets all outstanding technical requirements.

Kind regards,

Gianpaolo Reboldi, MD, MSc, PhD

Academic Editor

PLOS ONE
---

## [Editor Report · Acceptance letter]

17 Nov 2022

PONE-D-22-05315R2 

Hyporesponsiveness to erythropoiesis-stimulating agent in non-dialysis-dependent CKD patients: The BRIGHTEN study 

Dear Dr. Narita:

I'm pleased to inform you that your manuscript has been deemed suitable for publication in PLOS ONE. Congratulations! Your manuscript is now with our production department. 

Kind regards, 

on behalf of

Prof Gianpaolo Reboldi 

Academic Editor

PLOS ONE